# Heat and Mass Transfer Kinetics on the Chemical and Sensory Quality of Arabica Coffee Beans

Danieli Grancieri Debona [1], Renata Falqueto Louvem [2], José Maria Rodrigues da Luz [3],
Yuri Nascimento Nariyoshi [2], Eustaquio Vinicius Ribeiro de Castro [1], Emanuele Catarina da Silva Oliveira [4],
Rogerio Carvalho Guarconi [5], Marina Gomes de Castro [4], Gustavo Falquetto de Oliveira [4], Fábio Luiz Partelli [6],
Marliane de Cássia Soares da Silva [3], Ademilson Pelengrino Bellon [4] and Lucas Louzada Pereira [4,*]

[1] Chemistry Department, Laboratory of Research and Development of Methodologies for Analysis of Oils,
   LabPetro, Federal University of Espírito Santo, Av. Fernando Ferrari, 514, Goiabeiras, Vitória 29075-910, Brazil
[2] Department of Engineering and Technology, Federal University of Espírito Santo, Rodovia BR 101 Norte,
   Km 60, São Mateus 29932-900, Brazil
[3] Departamento de Microbiologia, Laboratório de Associações Micorrizicas, Universidade Federal de Viçosa,
   Av. Ph Rolfs, S/N, Viçosa 36570-000, Brazil
[4] Federal Institute of Education Science and Technology of Espírito Santo, Coffee Design Group, Rua Elizabete
   Minete, 500, Venda Nova do Imigrante 29375-000, Brazil
[5] Capixaba Institute for Research, Technical Assistance and Rural Extension, BR 262 km 94,
   Domingos Martins 29278-000, Brazil
[6] Department of Agricultural and Biological Sciences, Federal University of Espírito Santo,
   Vitória 29932-900, Brazil
*   Correspondence: lucas.pereira@ifes.edu.br or lucaslozada@hotmail.com

**Abstract:** Roasting has been used by the coffee industry to promote changes in the physical and chemical structure of coffee beans that influence the sensory quality of coffee beverages. However, there are no standardization rules for the temperature and roasting time. Thus, this study evaluated the influence of four roasting profiles obtained by two different roasters on the chemical and sensory quality of the coffee bean. Baked, light, medium, and dark roasting were evaluated using medium infrared spectroscopy and cupping test. Individual and joint effects of temperature and time for each roasting profile were observed on the loss of grain mass. There are specific regions in the infrared spectrum that can be used as markers to discriminate the roasting profiles and the type of roaster used. Despite the difference observed in the ranges of the infrared spectra, the roasters did not present significant differences in the average of the final sensory notes. This result shows the need to use analytical chemical techniques together with sensory analysis in order to better determine differences between coffee samples. Therefore, differences observed in the chemical analyzes and in the sensory attributes of roasted coffee are related to the roasting profile and type of roaster.

**Keywords:** thermal radiation; convection; conduction; coffee beverage; physical changes; Brazilian coffee

## 1. Introduction

Roasting is performed by applying heat energy to a material for a period. This process has been used by the coffee industry to promote changes in the physical, chemical, and sensory structure of the beans [1,2].

Different roasting profiles (baked, light, medium, and dark) may be observed during the coffee bean roasting process [1,3]. These roast profiles are established by monitoring time and temperature during roasting [4]. However, there are no standardization rules for such variables that are directly influenced by the thermal matrix (conduction, convection, and radiation) and by the roasters [5]. Furthermore, the roasters have manufacturer-specific engineering.

The use of different thermal matrices in the development of roasters promotes changes in heat transfer during roasting [6]. These changes influence the roasting profiles, physical structure, and chemical composition of coffee beans [7,8].

Heat transfer in roasters can be through thermal radiation, convection, and conduction. Radiation is little used compared to other thermal matrices [2]. In thermal radiation, heat transfer propagates by electromagnetic waves in a vacuum at the speed of light, therefore, unlike conduction and convection, it does not need a material medium for the energy transfer to take place [9]. Convection occurs in fluids where energy is transferred due to mass motion and the temperature gradient in space [9]. Thus, convection roasters have an air flow. Conduction occurs via the collisions between atoms and molecules of a substance and the subsequent transfer of kinetic energy. During roasting, thermal conduction is characterized by the transfer of thermal energy from the heated walls of the roasting chamber to the beans [10]. The conduction system is most common among the three types of heat transfer, but cylindrical drum roasters that have an integrated manual airflow control system feature the three thermal matrices. Heat transfer occurs mainly through conduction and convection, but also through radiation that participates in a limited way through absorbed, reflected, and transmitted radiation to the grains [2,11].

The efficiency of the roasting process and the sensory quality of the roasted coffee depend on physicochemical parameters, such as chemical composition, temperature, pressure, gas flow rate, relationship between time and relative speed of the beans, speed drum rotation, air flow rate and speed, maximum and minimum drum capacity, and the composition of the material used in making the roaster [7].

Each roaster needs proper adjustments of these parameters, as they affect the rate of heat transfer to the coffee bean and chemical reactions induced by this heat, such as the Mallaird and Caramelization reactions [7]. These chemical reactions produce volatile compounds that characterize the sensory attributes and the final quality of the coffee beverage.

The sensory quality of the coffee beverage has been evaluated by the method of the Specialty Coffee Association of America [12]. This sensory method is based on a protocol used by panelists (Q-Graders) to score ten sensory attributes (e.g., fragrance/aroma, uniformity, absence of defects, sweetness, flavor, aftertaste, acidity, body, and balance). However, the SCA protocol has been questioned due to the subjectivity of the Q-Grades descriptions and the variations in the sample preparation (water temperature, coffee powder mass, mill granulometry) that affect the repeatability of the process and the evaluation [13]. Thus, the use of analytical techniques would be an alternative to complement the results obtained by the sensory method [3].

Infrared spectroscopy is a fast and low-cost analytical technique capable of investigating the structural composition of molecules and predicting the functional groups of the sample [14], based on vibrational changes or stretching of their chemical bonds. This technique may present some chemical markers of coffee that are related to its sensory characteristics obtained by the SCA protocol. Thus, infrared spectroscopy becomes an ally in defining the sensory and chemical quality of coffees from different roasting processes [15].

Thus, this study evaluated the influence of four roasting profiles (baked, light, medium, and dark) obtained by two different roasters on the chemical and sensory quality of roasted coffee beans.

## 2. Materials and Methods

### 2.1. Samples

The Arabica coffee beans were obtained from the mountain region of Espírito Santo, Brazil. Only coffee beans with 85% of maturity were manually harvested at 1050 m of altitude. After that, an quantity of 120 kg of beans were washed with water and then classified as green, float, or mature in the Laboratory of Analysis and Research in Coffee at the Federal Institute of Espírito Santo, Brazil–Campus Venda Nova do Imigrante.

The mature beans (cherry coffee) were taken to a peeling machine model DRC-2 (Pinhalense®, Espírito Santo do Pinhal, Espírito Santo, Brazil) and then directed to a semi-

dry processing on a suspended yard until reaching 12% of humidity [16]. Finally, the beans were classified by sieving (Standard Sieves, Tyler Series) and the defective beans that passed through a 16-mesh sieve (6 mm opening) were discarded.

### 2.2. Roasting Procedures

The assays were carried out using two different drum roasters (Table 1, Figure 1) in batches of 1.2 or 2.0 kg samples under well-defined process conditions. The effect of continuous temperature increase on both roasters was investigated by applying different heating rates (Figure 2) starting at 150 °C. We performed three samples per roasting profile and roasters. Bean roasting was monitored by on-line temperature measurements. In order to compare the experiments, roasting assays were controlled to achieve four different degrees of roast determined by comparison with color disks from the "Roast Color Classification System" developed by Agtron e SCA [17,18]; thus, the baked roast presented a coloring degree equal to 80, light roasting a coloring degree of 65, medium roasting a coloring degree equal to 68, and the dark roasting a coloring degree equal to 56. The experiments were finalized as soon as the coffee beans achieved their reference color and then stored in an air-tight container for subsequent analyses. Colors of the coffee beans in each roasting profile can be observed in Figure 2 from Anastácio et al. [1].

**Table 1.** Roaster's specifications given by the suppliers.

| Characteristics | Probat®-Probatino * | Atilla®-2 kg Gold Plus ** |
|---|---|---|
| Length (m) | 1.13 | 1.35 |
| Width (m) | 0.56 | 0.75 |
| Height (m) | 0.81 | 1.90 |
| Weight (kg) | 115 | 180 |
| Fuel type | Liquefied Petroleum Gas (LPG) | LPG |
| Power (kw) | 0.55 | 0.75 |
| Capacity (kg) | 0.8 to 1.2 | 2.0 |
| Roasting time (min) | 8 to 15 | 15 |
| Drum material | Carbon steel | Carbon steel |
| Drum wall type | Double | Single |
| Drum wall thickness (mm) | Not informed | 2 |
| Drum wall diameter (mm) | 200 (inner) | 285 |
| Drum wall lenght (mm) | 300 (inner) | 200 |
| Thermal insulation | Ceramic fiber | Absent |

* [19]; ** [20].

### 2.3. Medium Infrared (MID) Spectroscopy

MID Spectra of the roasted beans were recorded on a Cary 630 FTIR Spectrometer (Agilent Technologies, Santa Clara, CA, USA) equipped with a diamond Attenuated Total Reflectance (ATR) accessory. The spectra were registered collecting 16 scans in the interval 650–4000 cm$^{-1}$ in order to detect organic compounds [21,22] at a nominal resolution of 4 cm$^{-1}$. The samples were ground separately in triplicate using a laboratory mill model Sette 270 (Baratza, Bellevue, WA, USA) adjusted to fine granulometry. Thus, the obtained powder was then compacted using a hand-operated pellet press model Press Pike 161-1100 (Pike Technologies, Fitchburg, MA, USA) without adding potassium bromide (KBr), as proposed by [23]. The storage and transport of the coffee samples (~0.1 kg) from the roasters to the mill and from the mill to the infrared spectrophotometer was carried out in a glass desiccator.

### 2.4. Cupping of Sensory Evaluation

The sensory evaluation was carried out 24 h after roasting of the coffee beans by the cupping test with 5 Q-graders [12]. These panelists are professionals trained and qualified by the Coffee Quality Institute [24].

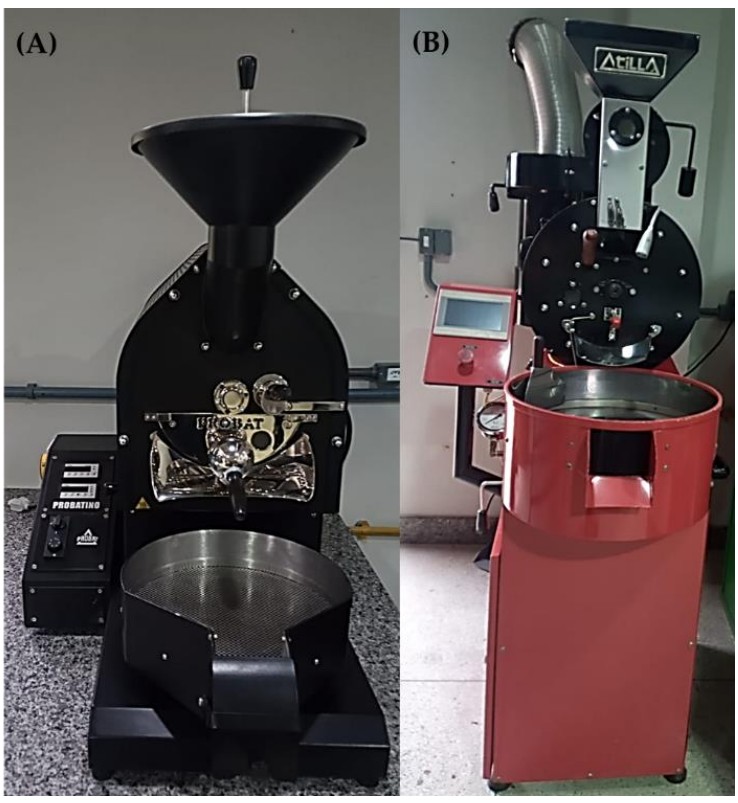

**Figure 1.** Probatino (**A**) (Probat, Emmerich am Rhein, Germany) and Golden Plus (**B**) (Atilla, Belo Horizonte, Brazil) roasters used in this study.

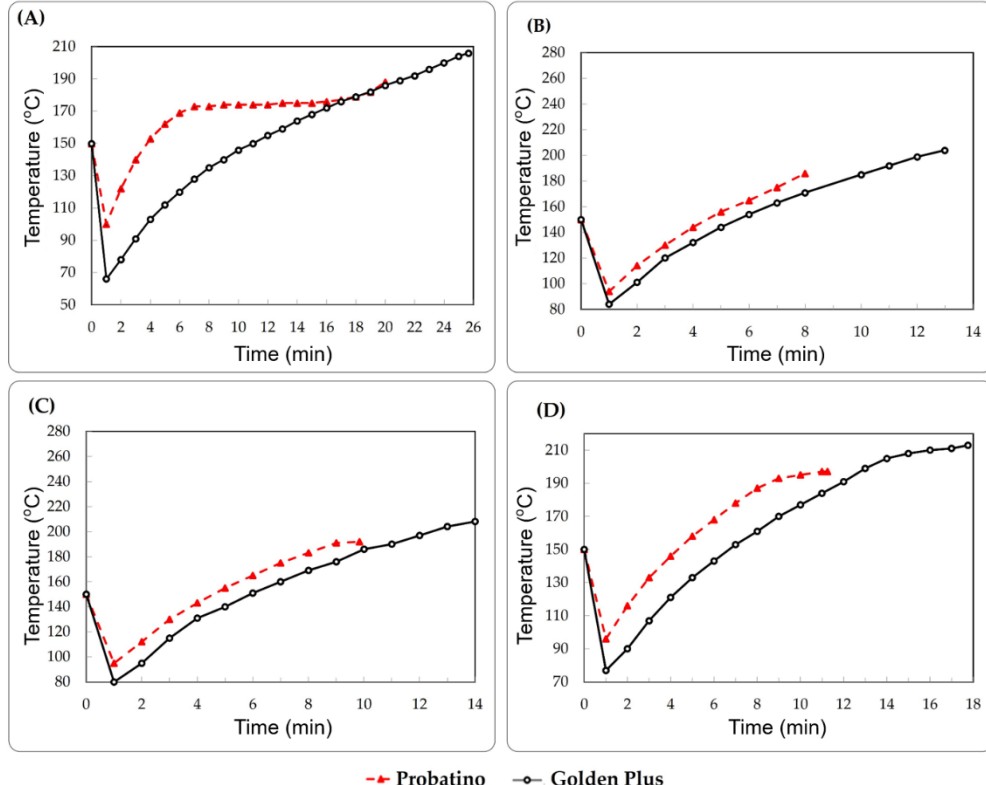

**Figure 2.** Baked (**A**), light (**B**), medium (**C**), and dark (**D**) roasting curves of coffee beans obtained from Probatino (Probat, Germany) and Golden Plus (Atilla, Brazil) roasters.

The coffee beans were ground by model Electric Mill G3 (Bunn Coffee Mill, Springfield, IL, USA) to medium/coarse granulometry, sieved through 20-mesh sieve (Tyler Series), and placed in five empty cups. Close to the coffee powder, the Q-Graders sensed the dry aroma, tasty and clean, and then poured hot water between 92 and 94 °C into the cups. Coffee powder with hot water ratio was 5.5% m/v, 8.25 g of coffee in 150 mL of water for each test. The Q-Graders waited 4 min, with the nose close for sniffing the aroma and recorded the smell, then stirred a little with a spoon, removed the coffee powder and foam from the surface, and then sipped to taste the coffee. In 12 min, as soon as the temperature reached 55 °C, to finish the cupping tests, the Q-Graders made sure the coffee came into contact with every taste bud in the mouth. Each sample of different roasters and roasting degree (Table 2) was tested, five cups at the same time.

**Table 2.** Roasting conditions for both coffee roasters to achieve four degrees of roast.

| | | Atilla®-2 Kg Gold Plus | | | | Probat®-Probatino | | | |
|---|---|---|---|---|---|---|---|---|---|
| Roasting Degrees | [1] Disk Scores | Roasting Time (min) | Heating Rate (°C·min$^{-1}$) | Final Temp (°C) | Weight Loss (%) | Roasting Time (min) | Heating Rate (°C·min$^{-1}$) | Final Temp (°C) | Weight Loss (%) |
| Baked | 80 | 25.67 | 6 | 206 | 9.8 b B | 20.00 | 5 | 188 | 12.7 a A |
| Light | 65 | 12.98 | 10 | 204 | 8.0 b B | 8.67 | 12 | 190 | 7.0 c A |
| Medium | 68 | 14.33 | 9 | 208 | 9.8 b A | 9.83 | 11 | 192 | 10.0 b A |
| Dark | 56 | 17.75 | 8 | 213 | 11.7 a A | 11.25 | 9 | 197 | 12.3 a A |

[1] The roast color was chosen according to the scale of the Agtron Disks (Agtron Inc., Reno, NV, USA). The uppercase letters in the rows and lowercase letters in the weight loss columns when equal show that there was no significant difference at the 5% level by Tukey's test.

### 2.5. Statistical Analysis

Roasting experiments were carried out in a completely randomized design, in a $2 \times 4$ factorial scheme (2 roasters with 4 degrees of roast each) with five replications. For the sensory results, we made use of the software R core, in which the experimental data were subjected to a variance analysis and the average values compared by the Tukey test at 5% of probability. For the evaluation of similarity between points of roaster/roasting profile, a matrix with average values of the variables and a dendrogram using the average euclidean distance to measure the distances between the coffees and the full link hierarchical grouping method were made. Finally, the MID spectra data were compiled into a matrix, in which each repetition was considered another sample. All the calculations were performed using the software Matlab version 2013 and Microsoft Excel version 2016. Before applying chemometric tools to the data set, the multiplicative scatter correction (MSC) was used as a data processing resource. In order to detect similarities and differences between the roasting profiles, the data were submitted to exploratory analysis using the Interval Principal Component Analysis (iPCA) technique.

## 3. Results and Discussion

### 3.1. Roasting Profiles

Roasting parameters were different between roasters regardless of roasting profile (Table 2). The final roasting time and temperature was higher in Golden Plus than in Probatino. Moreover, grain mass loss was higher in Probatino than Golden Plus for baked, medium, and dark roast profiles. This difference observed in these roasting parameters could be due to the technical specifications of each roaster (Table 1), which requires different handling by the operators, and the variations and thermal amplitude of the process. In addition, the optimal time and temperature for each coffee roasting profile was determined by the skill of the equipment operator, which causes differences in the chemical and sensory analysis of the roasted coffee. The development of prediction models using machine learning has been an interesting alternative [25]. Thus, the study of chemical compounds in roasted coffee can help in this development.

A thermal variation of up to 50 °C between the initial and final roasting temperature in Probatino was observed in a shorter roasting time than in Golden Plus (Figure 2, Table 2). The greater loss of grain mass in Probatino may be due to this thermal variation. In the light roasting profile, where the smallest variation of initial and final temperature was observed in Probatino, the smallest loss of mass was also observed when compared with other roasting profiles of this roaster. According to Bustos-Vanegas et al. [26], the roasting temperature is the main parameter that determines the kinetics of mass loss of coffees during roasting. In the light roasting profile, the temperatures and roasting time are lower than the values observed in the other roasting profiles. Thus, the lower mass loss observed in the light profile (Table 2) may be due to a shorter exposure time to high temperatures, as also observed in the study by Pereira and Moreira [10].

In the average roasting profile, there was no significant difference in mass loss between roasters, although a higher temperature increase and a shorter roasting time in Probatino compared to Golden Plus was observed. Golden Plus had a longer roast with a higher final temperature; this was due to the temperature variation having more homogeneous increments (Figure 2, Table 2). Thus, the mass loss occurred more slowly and was proportional to the roasting time (Table 2). The growth rate was different between roasters, which shows the influence of roasting time on the mass loss of coffee beans in each roasting profile. Anastácio et al. [1] also observed loss of mass/density as a function of roasting time in coffees subjected to different roasting profiles using the Type 2 Probatino roaster. Furthermore, the evaporation of water and the release of gases promote a loss of density of the coffee bean that is directly proportional to the roasting time [27].

The roasting kinetics of coffee beans can occur individually and together; the effects of temperature and time for each roasting profile is shown Table 2. The optimization of the effects of roasting time on the thermal contaminants of roasted coffee beans (e.g., 5-hydroxymethylfurfural, acrylamide, furan, 2-methyl furan, and 3-methyl furan) showed that the increase in mass loss and the decrease in the moisture contents of the beans are observed in first-order kinetics [28]. Physical changes in coffee bean structures as a function of roasting temperature influence the sensory attributes of coffee [1]. The concentration of 5-methylfuran reduced with increasing temperature and roasting time in robust coffee samples [29]. Coffee roast profiles feature different mass loss thermogravimetric curves for each roast profile [30]. According to these authors, the thermal degradation of the coffee chemical ingredients in the roasting profiles (light and dark roasted) starts in the temperature range of 171–188 °C. Therefore, roasting time and temperature are important variables for the chemical and sensory quality of coffee. Furthermore, sensory quality and lexical emotions (e.g., quiet, relieved, satisfied, merry, nostalgic, and off-balance) after the consumption of a cup of coffee were dependent on the roast profile [31]. This quality is directly related to the chemical composition of the roasted coffee [1,3,16].

### 3.2. Chemical Profiles

To prepare the iPCA, the infrared spectra obtained were divided into 15 wavelength bands (Figure 3). This division into intervals was based on the absorption and vibration bands characteristic of an infrared spectrum to compare the chemical composition of coffees roasted in two roasters with different roasting profiles.

The intervals that most contributed to a separation trend between Golden Plus and Probatino roasters were IV and XII (Figures 3 and 4). In these wavelength ranges, there is a tendency to separate roasters along the third principal component (PC3) for interval IV (Figures 3 and 4A) and in PC2 for interval XII (Figures 3 and 4B). These data are important to correlate the spectral bands with the changes in the chemical composition of the roasted coffee, because according to Alessandrini et al. [32], the formation or degradation of compounds affect the final spectral profile of roasted coffee.

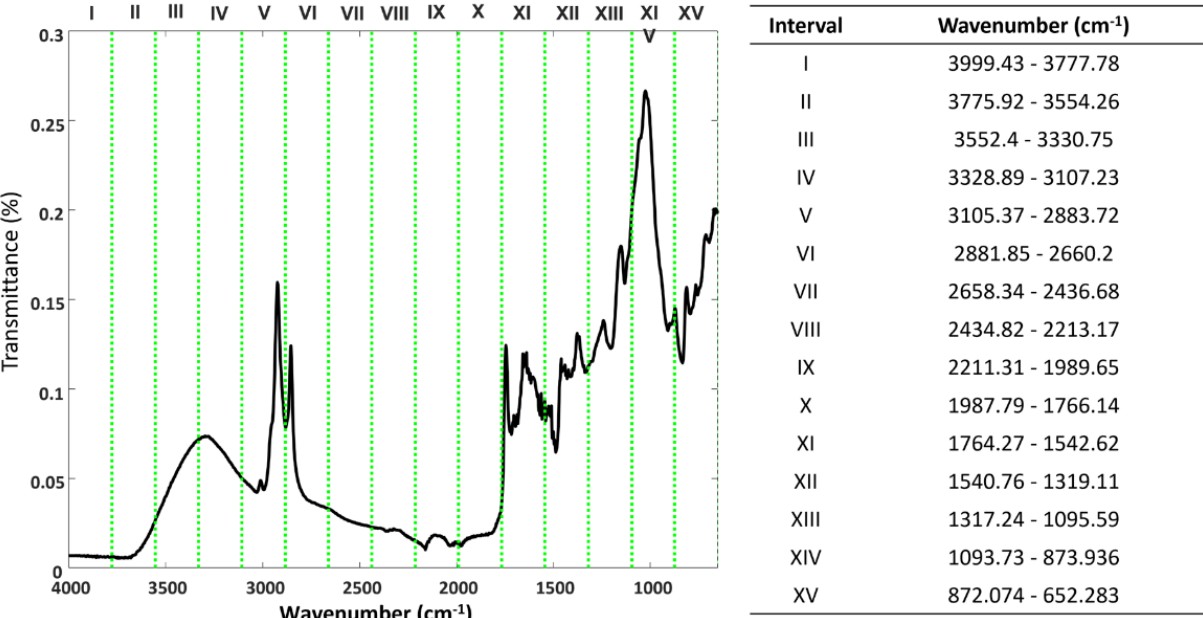

**Figure 3.** Infrared spectrum obtained from coffee samples roasted in different roasting profiles (baked, light, medium, and dark) using two roasters (Probatino and Golden Plus). This spectrum was divided into 15 intervals (I to XV) for application of the Interval Principal Component Analysis (iPCA) technique.

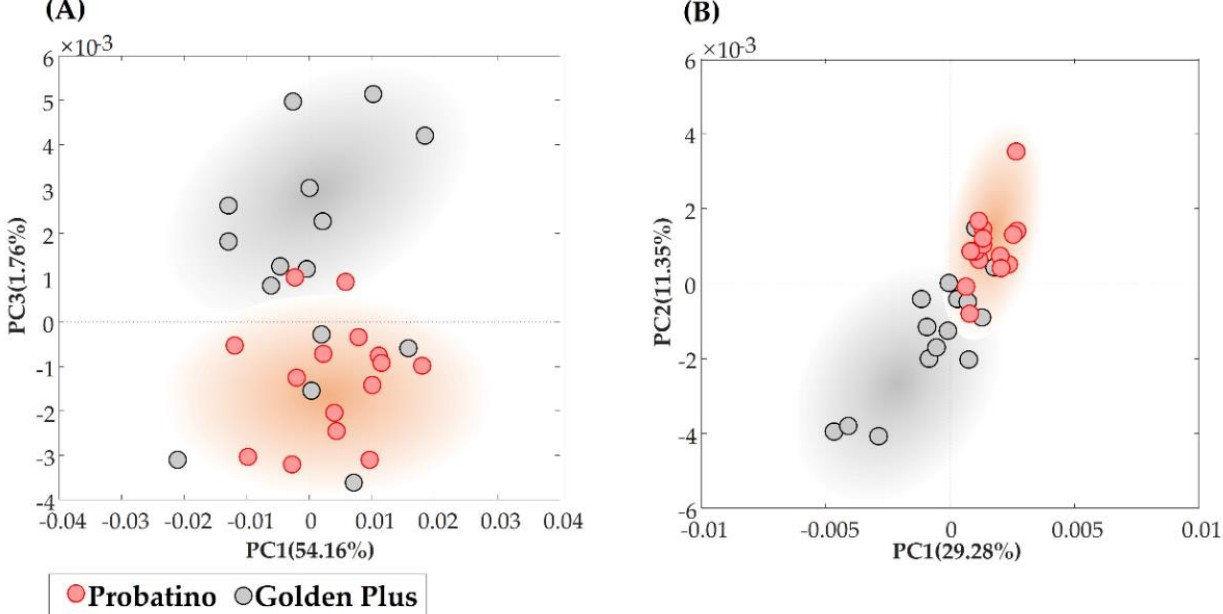

**Figure 4.** Interval Principal Component Analysis (iPCA) of the infrared spectra in the IR (**A**) and XII (**B**) intervals, as shown in Figure 3. These spectra are from samples of coffee roasted in different roasting profiles (baked, light, medium, and dark) using two roasters (Probatino and Golden Plus).

In the XII interval (Figures 3 and 4B), the spectral differences may be due to the stretching of the N-H bonds that characterize amines and secondary amides [21] and the axial deformation of C=C and C=N bonds. Caffeine is an alkaloid with absorbing bands in the region of 1650–1600 cm$^{-1}$ [33,34]. The trigonelline that is present in raw and roasted coffee has absorption bands in the range of 1650–1400 cm$^{-1}$ [35,36]. The chlorogenic acids (caffeic acid ester, quinic acid, p-coumaric, and ferulic acid) in coffee show strong absorption bands at 1450–1000 cm$^{-1}$ [37]. In addition, the separation of coffee samples

by PC3 (Figure 4A) is important, as the XII range of the spectrum presents compounds, for example, caffeine, chlorogenic acids, and trigonelline, that contribute positively to the sensory attributes of the coffee beverage.

The spectral bands present in the IV range (Figure 3) tend to be intense and sharp and are characteristic of stretching of N-H and O-H bonds. According to Reis et al. [35], these bands are common in roasted coffee samples, so the higher number of significant spectral differences in this band could be due to primary amines of amino acids and sugars. These compounds are precursors of coffee aroma and flavor and contribute to the occurrence of the Maillard reaction and Strecker degradation [21,38,39]. These thermal reactions also produce compounds responsible for the sensory characteristics of roasted coffee [38,39].

A separation between the roasting profiles for both roasters was also observed in the iPCA of the infrared spectra (Figures 5 and 6). Two groups with Golden Plus light, medium, and dark roast profiles were observed in PC1 and PC2 using the spectrum intervals IV and XIII (Figures 3 and 4A,B). In the range of XIII (Figure 3), bands of the C-O bonds of carboxylic acids, esters, ethers, alcohols, and phenols can be observed in the spectra [21]. Organic acids, esters, and phenols are found in roasted coffee samples and in the beverage [34].

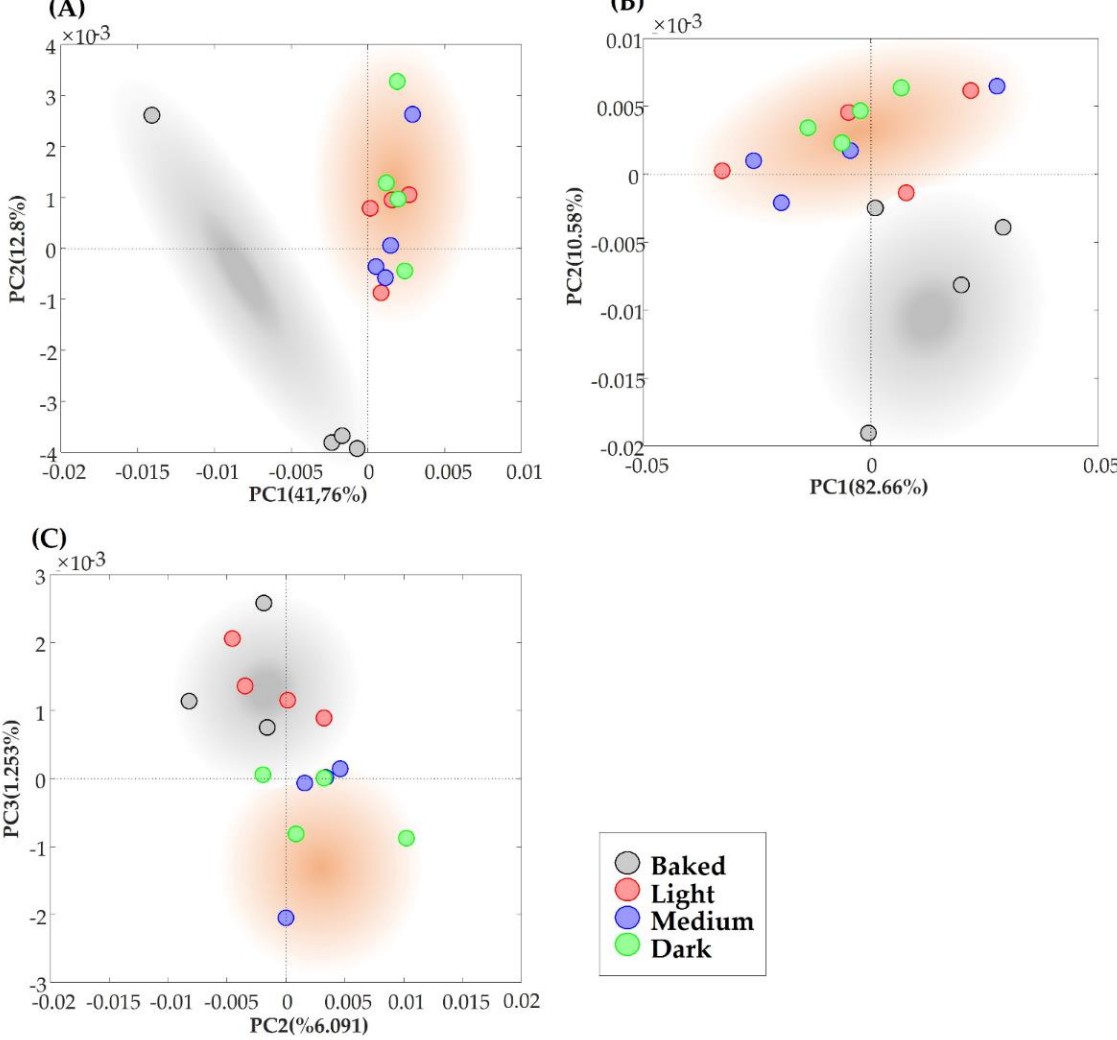

**Figure 5.** Interval Principal Component Analysis (iPCA) of infrared spectra in the IV (**A**), XIII (**B**), and V (**C**) intervals, as shown in Figure 3. These spectra are from samples of coffee roasted in different roasting profiles (baked, light, medium, and dark) using two roasters (Probatino and Golden Plus).

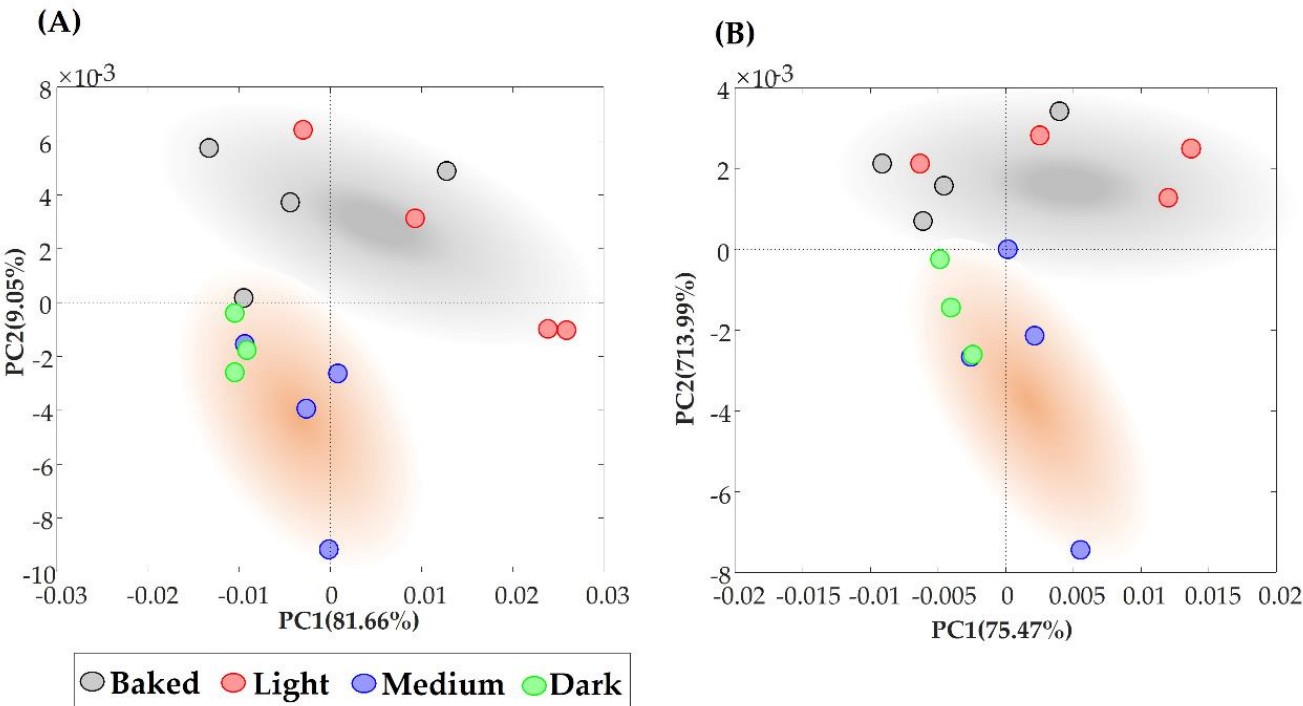

**Figure 6.** Interval Principal Component Analysis (iPCA) of the infrared spectra in the V (**A**) and VI (**B**) intervals, as shown in Figure 3. These spectra are from samples of coffee roasted in different roasting profiles (baked, light, medium, and dark) using two roasters (Probatino and Golden Plus).

In the spectral range of the V range, there was also a separation in the PC3, where two groups were formed by the Golden Plus roast profiles (Figures 3 and 4C). The first group formed by the baked and light roast profile scores are distributed in PC3 > 0 and the second group in PC3 < 0 (Figure 4C). In this range, there is the presence of C-H bonds of methyl groups, present in caffeine [37], inferring that it was responsible for the separation, since roasts with higher temperatures and degree of roasting tend to present greater caffeine content when compared in mass proportion with the other compounds that degrade throughout the roast because they are not thermally stable [40].

In the Probatino roaster, the roasting profiles tend to separate along the PC2 for the V and VI intervals of the infrared spectrum (Figures 3 and 6A,B). Two groups were observed at these intervals for the roasting profiles (Figure 6A,B). The first group with the light and baked roast profiles at PC2 > 0 and the second group with the medium and dark roast profiles PC2 < 0 at both intervals. Thus, the light and baked profiles present similar chemical classes, but different from those of the medium and dark profiles. Furthermore, the medium and dark profiles also show similarity in relation to the chemical classes of the infrared spectrum.

The spectral bands of the range V and IV (Figures 3 and 6) refer to the C-H bond of the methyl group (3000–2800 cm$^{-1}$) present in the caffeine molecule [37,41]. However, this bond, due to its thermal stability, is not considered a roasting marker [42]. In these intervals, the stretching vibration of the H-C= bonds is also observed, which may be related to unsaturation in fatty acids and to the antisymmetric and symmetrical vibrations of methyl [43]. There is also an indication of aldehydes that are formed from the degradation of hydroperoxides and other unsaturated chains during roasting [44], in the band range 2860–2800 cm$^{-1}$ (Figure 3).

### 3.3. Sensory Profiles

Despite the difference observed in the iPCA of the infrared spectra, the roasters did not show significant differences ($p > 0.05$) in the average of the final sensory note (Table 3). This result shows the need to use analytical chemical techniques together with

sensory analysis in order to better determine the differences between coffee samples. According to Debona et al. [3], the use of instrumental and qualitative methods in association with sensory analysis is advantageous to safely assess the quality of the coffee beverage. In addition, the chemical results and the separations presented in the iPCA in the ranges 3328.89–3107.23 cm$^{-1}$ and 1317.24–1095.59 cm$^{-1}$ for the Golden Plus roaster (Figures 3 and 5) corroborate the sensory results (Table 3, Figure 7). In this analysis, the coffee with the baked roast profile showed different levels of similarity regarding the sensorial quality of the coffee in relation to the other roast profile (Figure 7).

**Table 3.** Final sensory note of samples of coffee roasted in different roasting profiles (baked, light, medium, and dark) using two roasters.

| Roasting Profiles | Roasters | | | | | | Average [1] | |
|---|---|---|---|---|---|---|---|---|
| | Golden Plus | | | Probatino | | | | |
| Baked | 79.40 ± 2.43 | ab | A | 79.35 ± 2.96 | a | A | 79.38 ± 2.69 | a |
| Light | 80.60 ± 2.04 | ab | A | 82.10 ± 2.79 | a | A | 81.35 ± 2.41 | a |
| Medium | 82.70 ± 3.58 | a | A | 81.90 ± 1.24 | a | A | 82.30 ± 2.41 | a |
| Dark | 78.00 ± 2.76 | b | A | 81.20 ± 2.14 | a | A | 79.60 ± 2.45 | a |
| Average | 80.18 ± 2.7 | A | | 81.14 ± 2.28 | A | | | |

[1] Average followed by at least one capital letter horizontally and at least one lowercase letter vertically does not differ from each other by Tukey's test at 5% probability.

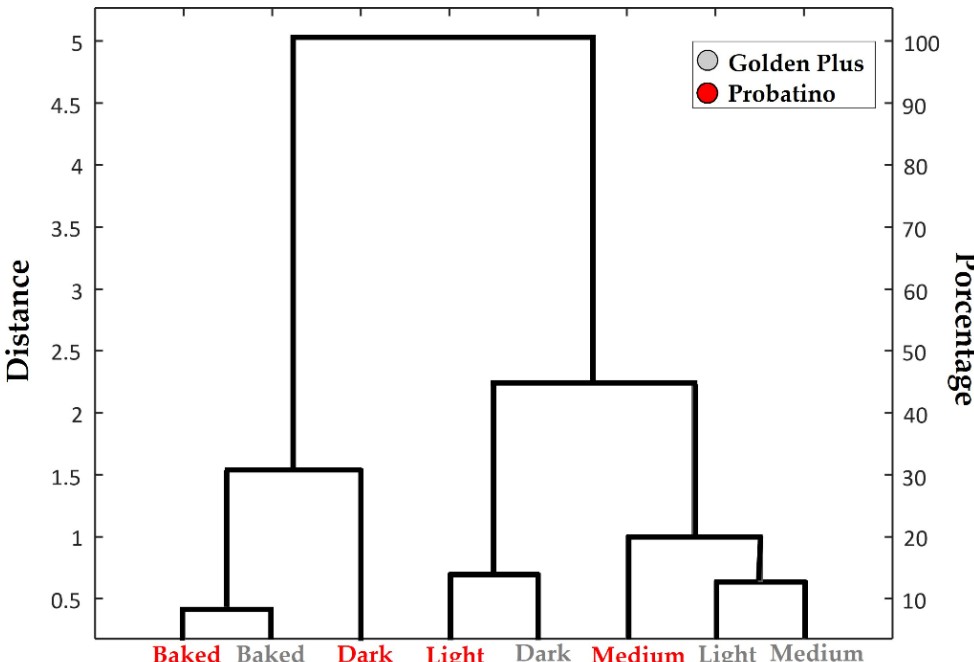

**Figure 7.** Dendrogram of the eight-point clusters (4 roasting profiles and 2 roasters) in relation to the sensory analysis of roasted coffee samples.

In the Probatino roaster, there was no significant difference ($p > 0.05$) in the final grades of the different roasting profiles (Table 3). However, the final grade of the medium roast was higher than the final grade of the dark roast, and significant differences were not observed between the other roast profiles of the Golden Plus roaster. According to Abubakar et al. [45], the degree of roasting is inversely proportional to the sensory note. In the dark level, the degradation of chemical compounds that are precursors of the formation of flavor and aroma in the coffee beverage occurs [3,40,45].

The dendrogram of roasters/roasting profiles and the sensory analyzes show the existence of two homogeneous groups with a similarity greater than 30% (Figure 7). Group

A was formed by the roaster/roasting profile points that had a final sensory note below 80 points (Table 3). According to Giacalone et al. [46], baked and dark roasts tend to have a lower sensory note than other roasting profiles. The rate of compounds formation not beneficial to the sensory quality of the coffee beverage (e.g., maltol, difurfuryl ether, and pyridine) is directly proportional to the coffee roasting time/temperature [47].

Group B consisted of light and medium roasts from the Golden Plus roaster and light, medium, and dark roasts from the Probatino roasters that had the best sensory scores (Figure 7, Table 3). The medium roasts had positive evaluations regarding the intensity of the sensorial attributes and the concentration of aromatic compounds [46]. Light and medium roasts have roast curves (Figure 2) and similar sensory notes. However, in light roasts, there is a reduction in the compounds formation due to the shortening of the roasting time [46]. This reduction in the formation of compounds may have a negative impact on the formation of aromas and flavors, however, no significant differences ($p < 0.05$) were observed in the sensory evaluation between these two roasting profiles (Figure 7, Table 3). These results confirm the importance of associating sensory techniques with analytical chemistry techniques, such as infrared spectrophotometry, to evaluate coffee quality.

The sensory attributes that most contributed to the construction of the roast profile dendrogram versus sensory attributes (Figure 7) were aftertaste (50%), balance (17.9%), fragrance (14.3%), overall (14.3%), and the flavor (3.5%). Group formation using these sensory attributes was also observed in the sensory discrimination of coffees from different planting altitudes [48]. The aftertaste is characterized by the persistence of the flavor on the palate, which may or may not be pleasant [12]. Balance is the synergy between flavor, finish, acidity, and body [12]. Fragrance and overall are aromatic characteristics [16]. The flavor is the main sensorial characteristic of the coffee, being considered the "central phase" of the evaluation that is between the first impressions [3,12]. The harmonization of these sensory attributes make up the sensory notes assigned by the Q-Grades (Table 3) and depends on the chemical composition of the roasted coffee [10,49].

## 4. Conclusions

Roasting time and temperature, independently of the roaster, are important variables for the chemical and sensory quality of coffee. Competing effects between these variables in relation to the mass loss of coffee beans may occur. The differences in the chemical and sensory analysis of roasted coffee are related to the roasting profile and the type of roaster used. There are specific regions in the infrared spectrum that may be used as markers to discriminate between roasting profiles (3328.89–3107.23 cm$^{-1}$ and 1093.73–873.93 cm$^{-1}$) and the type of roaster used (3328.89–3107.23 cm$^{-1}$ and 1540.76–1319.11 cm$^{-1}$).

**Author Contributions:** Conceptualization, D.G.D., L.L.P., R.F.L., M.G.d.C., G.F.d.O., M.d.C.S.d.S. and J.M.R.d.L.; methodology L.L.P., E.C.d.s.O. and R.C.G.; validation E.V.R.d.C., M.G.d.C. and G.F.d.O.; formal analysis E.C.d.s.O., R.C.G., R.F.L. and D.G.D.; investigation J.M.R.d.L., L.L.P., F.L.P., Y.N.N. and D.G.D.; data curation Y.N.N., R.C.G. and E.C.d.s.O.; writing—preparation of the original draft M.d.C.S.d.S., J.M.R.d.L. and D.G.D.; writing—review and edition D.G.D., J.M.R.d.L., M.d.C.S.d.S., A.P.B., Y.N.N. and R.F.L.; visualization E.V.R.d.C. and J.M.R.d.L.; supervision L.L.P. and J.M.R.d.L. All authors have read and agreed to the published version of the manuscript.

**Funding:** This research was funded by Free Admission Credit Cooperative—Sicoob grant number 23186000886201801.

**Data Availability Statement:** Not applicable.

**Acknowledgments:** The authors would like to thank Sul Serrana of Espírito Santo Free Admission Credit Cooperative—Sicoob (23186000886201801), CAPES (Coordenação de Aperfeiçoamento de Pessoal de Nível Superior—Código de Financiamento 001), CNPq (Conselho Nacional de Desenvolvimento Científico e Tecnologia), and Instituto Federal do Espírito Santo, for supporting the research, through the PRPPG no. 12/2021—Researcher in Productivity Program–PPP and the Q-Graders for their cooperation in this study. We are also very thankful to the Laboratory of Research and Development of Methodologies for Analysis of Oils-Federal University of Espírito Santo (LABPETRO/UFES)

and Cropster Brasil for donating the license for Cropster software. In addition, we also extend our thanks to the coffee growers for their donation of the coffee cherries, which allowed for the development of this study.

**Conflicts of Interest:** The authors declare no conflict of interest.

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
