# Peer review of "Heat and Mass Transfer Kinetics on the Chemical and Sensory Quality of Arabica Coffee Beans"

_agronomy, doi:10.3390/agronomy12112880_

Round 1
Reviewer 1 Report
The study evaluated the influence of four roasting profiles obtained by two different roasters on the chemical and sensory quality of roasted coffee beans. The authors should give more explanation about the sensory properties of different coffee samples. Table 3 seems to be the sensory note of coffee samples, But I am wondering how do the experiments conduct? Usually, the sensory tests could collect the sensory profiles of evaluated samples on varied attributes.
Author Response
Dear,
Thank you very much for your comments and suggestions on the manuscript. We have modified the manuscript accordingly, and the detailed corrections are attached to the document point by point. In manuscript, we corrections are in red.
Thank you for your collaboration

Reviewer 2 Report
Dear Authors,
this is a very interesting publication and the research methods used in it.
I have one remark regarding the measurement units and their notation in Table 1 and Figure 1. Please convert: (Kg) to (kg), (Kw) to (kW) and (Min) to (min).
These are minor editorial errors which do not diminish the high substantive value of the work.
Yours sincerely
Piotr Sałek, PhD
Author Response
Dear all,
Thank you very much for your comments and suggestions on the manuscript. We have modified the manuscript accordingly, and the detailed corrections are attached to the document point by point. In manuscript, we corrections are in red.

Reviewer 3 Report
The topic of impact of roasting profiles and roaster type on the chemical and sensory quality of arabica coffee beans is very interesting, which has great practical reference value for the coffee industrial. However, the present form is far way to provide convincing scientific answer to this topic. Major revision is recommended. Please find my suggestion for authors consideration.
1. Infrared is often used for qualitative study to show the variation of the bond between the two atoms, and sensory cupping is a good way to show the overall perception of coffee. Both are good for giving an overall impression of change. But they are not enough to profile the impact of certain kind of change…A quantitative GC or LC study is strongly recommended to better illustrate how the roaster type would affect the coffee flavor and tasting profile from the compositional basis.
2. The author should consider providing pictures of roasters and how the roasted coffee bean look like, which would help the readers understand the machine and product difference.
3. For the roasting procedure, heat and mass transfer kinetics would be an interesting topic to give more insightful discussion, to reveal and connect this kinetic difference to the roaster type would better fit the paper title.
4. How did you prepared the sample before sending to MID-IR? What is the particle size of coffee beans?
5. How many panelists did you have for your cupping sensory? How many men and women, what were their ages?
6. Figure 2, please provide the y axis caption. With only one curve in the Figure, the readers cannot tell which one belongs to Probatino and which one belongs to Golden Plus. Please consider listing corresponding bond next to the wavenumber of in Table in Figure 2.
7. The authors tried very hard to differentiate different roasting degrees and different roasters in session 3.2, yet the cupping tasting results in session 3.3 did not show much significance among roasting degrees nor roasters… So could people tell the difference between different roasting degrees or different roasters in the daily basis? If yes, what kind of chemical play the critical role in tasting difference? If no, do we need that much effort to differentiate different roasters?
8. In the conclusion, with only one kind of bean used in this study, it might not be appropriate to state certain band regions could be use as markers to discriminate the roasting profile, considering the sample was too limited…
Author Response

(The authors gave the same response as above.)

Reviewer 4 Report
In this form, the article is just a comparison of two roasting roaster's. I miss other analyses such as the determination of volatiles by gas chromatography, or the determination of organic acids by liquid chromatography, or the determination of caffeine itself. The sensory analysis part of the analysis could be better handled and the difference in sensory profiles could be better displayed. Certainly this study should be complemented by other coffee varieties or different processing of coffee beans.
Author Response
Dear,
Thank you very much for your comments and suggestions on the manuscript. We have modified the manuscript accordingly, and the detailed corrections are attached to the document point by point. In manuscript, we corrections are in red.
Thank you for your collaboration.

Round 2
Reviewer 4 Report
- I don't have any comments